# Main drivers of regional value chains in CAFTA: Does trade facilitation matter?

**Yang Yu**[1,2], **Zhouying Song** [1,3]*

**1** Institute of Geographic Science and Natural Resource Research, Chinese Academy of Sciences, Beijing, China, **2** Greater Bay Area Big Data Research Institute, Shenzhen, China, **3** College of Resources and Environment, University of Chinese Academy of Sciences, Beijing, China

* songzy@igsnrr.ac.cn

## Abstract

In the last decades, economic globalisation and the progress of ICT have promoted the international division of labour and optimisation of the global value chain. Moreover, improvements in incentives such as lower tariffs and more efficient border crossings have boosted international trade. Under this background, regional and sub-regional economic cooperation organizations, such as free trade area (FTA), have been developing rapidly and attracting many academic attentions. As the fastest growing FTA in the world, CAFTA is now the largest FTA in developing countries. This study focuses on the value-added net-work of various industries in the trade process inside CAFTA, and tries to explore the impact of trade facilitation on the *DVA* trade network of CAFTA. The results show that in the trade network of CAFTA, the proportion of added value of domestic trade in total exports keeps increasing, and the returned added value (*RDV*) increases significantly. Singapore, Vietnam, and Thailand hold relatively high positions in the production network, while China has a relatively low position. On the other hand, China and Thailand become the main beneficiaries after the establishment of CAFTA, while Singapore and Malaysia play a lesser role in trading networks. The results also show that trade facilitation has a significant positive effect on the *DVA-INTrex* and *RDV* trade networks, indicating that trade facilitation can significantly increase the domestic indirect value added and returned value added in the trade process. Moreover, the business environment (*bus*) is the most important factor, with efficiency and transparency of border administration (*cus*), availability, and use of ICTs (*ict*) contributing to the improvement.

## 1. Introduction

With the rapid progress of information, communication, and transport technology, the value chain of products has been continuously subdivided into independent links. At the same time, a complete global value chain (GVC) has been formed by transferring each link to the country with the lowest cost [1]. Along with the increasingly detailed international division of labour in products of developed countries in different regions, some developing countries embed themselves into the global production network (GPN). In this process, regional value chains

---

**Data Availability Statement:** All relevant data are within the manuscript.

**Funding:** This research was funded by the Priority Research Program of Chinese Academy of Sciences (XDA20010102) and Research Program for Oversea Cooperation Center of Chinese

Academy of Sciences (162GJHZ2022004MI). The funders had no role in study design, data collection and analysis, decision to publish, or preparation of the manuscript.

**Competing interests:** The authors have declared that no competing interests exist.

(RVCs) play a vital role in boosting manufacturing production, given the fact that regional trade is dominated by trade in manufactured goods [2]. In the last decade, the share of intra-Asia trade in value-added is high at 45%, compared to 9% in Africa and 18% in Latin America [3]. In fact, China and the Association of South-east Asian Nations (ASEAN) have presently become an important choice for offshore outsourcing in Asia with the exception of India [4]. Furthermore, China and ASEAN have strengthened economic and trade cooperation since the 1990s, especially after the establishment of the China and ASEAN Free Trade Area (CAFTA) in 2010.

In the last decades, with the rapid development of economic globalization and international division of labour, regional and sub-regional economic cooperation organization, such as FTA, has been developing rapidly. In the 2010s, CAFTA has been the largest and fastest growing FTA among developing countries in the world. According to the United Nations Conference on Trade and Development (UNCTAD), the annual growth rate of CAFTA's total trade reached 11.36%, much higher than 5.50% in EU and 4.65% in NAFTA from 2001 to 2018. Inside CAFTA, China was ASEAN's largest trading partner and ASEAN was China's third largest trading partner in 2018. Moreover, China and ASEAN countries are large and increasingly diverse nations in terms of social, political, economic, and demographic attributes. For example, with approximately 4.62 trillion USD in trade volume (largest in the world) and almost 3.01 trillion USD domestic value-added in trade volume (second only to the United States) in 2018, China is among the top two nations worldwide in terms of international trade scale. Therefore, complicated CAFTA trade networks are to be expected.

Under the CAFTA framework, China and ASEAN countries have launched a series of facilitation measures to promote regional trade through tariff reduction, which make it easier for the intermediate products to cross the national borders many times in the process of trade. And the fast development of CAFTA has proved that the facilitation measures have a positive impact on transnational production division. Existing study also demonstrates that together with information and communication technology (ICT), trade facilitation measures could promote vertical specialisation division [5, 6]. Furthermore, trade facilitation levels have also been shown to have an obvious promoting effect, by which developing countries were stronger than developed countries in the improvement of product export by current studies [7]. However, how the trade facilitation measures inside CAFTA promote the regional trade cooperation is not clear.

In this study, we focus on the value-added in trade within CAFTA. The aims of this study are to develop a conceptual research framework to explore the value-added in trade network of CAFTA, to examine the RVCs in intra-CAFTA countries, and to identify the impacts of trade facilitation factors on the different kinds of value-added in trade networks of CAFTA. Additionally, based on the findings, we aimed to provide empirical evidence for China and Southeast Asian countries, and further provide political and management recommendations for developing countries.

This study has some novel features and makes three contributions to the international trade scholarship. First, it analysed the RVCs from the perspective of value-added in trade at the bilateral level, which was rare. It answers calls for multiple-level studies and thereby substantially extends comparative empirical examinations of international trade. For this, it used primary data on ADB-Mrio, using a scale of analysis in input-output table. Second, it systematically analysed the influences of facilitation factors on the value-added in trade, which is of great significance to reveal trade facilitation to the RVCs. Although some scholars have discussed the influence of facilitation factors on *Returned Domestic Value-added* (*RDV*) in trade, there has been no research comparing the impact of facilitation factors on different kinds of value-added in trade. Third, it enhanced the quality and rigor of the comparative

analysis by applying spatial analysis and mapping methods, which are significant supplements for traditional multivariate analysis. It employed the visualisation capabilities of ArcGIS to develop a descriptive understanding of the network of value-added in trade in CAFTA to show RVCs.

## 2. Research framework

### 2.1 Value-added in trade and the decomposition of export value

With the global division of labour and the rapid increase in trade in intermediate goods, the traditional international trade accounting data were considered to be distorted, and the actual export value of some processing trade-oriented countries was exaggerated. Measuring the value flow in the trade process has become an important research component of GVC [8].

Early scholars focused on vertical specialisation and value decomposition in trade processes. Hummels defined the concept of vertical specialisation as measuring a country's integration into GVCs by using the proportion of foreign value contained in exports within a country [9]. Through the *HIY* method [10], the total export of a country was summed up as domestic value added and foreign value added. And a model of export value decomposition was created. Moreover, specific indicators of vertical specialisation were studied in depth from the perspective of production and consumption in the later research [11]. In the 2000s, the definitions of "Value-Added in Trade" and "Trade in Value-Added" were well distinguished and became the main ways to measure the actual value flow in international trade [12].

And there are some research, which has achieved a preliminary decomposition of trade in value-added [13]. For example, based on the shortcomings of the traditional trade accounting system, Koopman et al. divided the export value of trade into three parts, domestic value added, circumvented value added, and foreign value added, with a total of nine indicators, and connected the decomposition method of added value trade with the traditional gross value trade accounting system effectively [14]. Wang et al. expanded the decomposition formula of the trade in value-added. Export value can be divided into domestic value-added eventually absorbed abroad (*VAX_G*), domestic value-added first exported then returned home (*RDV*), foreign value-added (*FVA*), and pure double counted terms (*PDC*) in four parts, for sixteen total indicators, with sixteen items in the Koopman decomposition method and nine strictly corresponding indicators [15]. Moreover, University of International Business and Economics (UIBE) further combined sixteen decomposition items according to the export value from the bilateral level and integrated them into eight decomposition items. The accounting framework is shown in Fig 1 as follows:

Based on the accounting framework (Fig 1), this study mainly focused on the domestic value-added. As the domestic value-added of a country's export is mainly used to reflect the actual interests of the country in international export. When we study the bilateral export process of CAFTA, we used the value-added decomposition method of *KWW* to calculate three parts of domestic value-added (*DVA*): the direct value-added (*DVA-D*), indirect value-added (*DVA-INTrex*), and re-import value-added (*RDV*). The meanings of various types of added value are shown in Table 1.

Consider the need for a multi-country or global input-output table through decomposition, it can be costly in terms of material and financial resources. At present, academia mainly calculates the value added of trade through the input-output table of many countries compiled by international organizations. For example, the OECD-WTO has built the TiVA database to calculate various trade added values of 40 major countries [16]. Based on the Input and Output Table of 187 countries, UNCTAD and the University of Sydney calculated the added value of each country's export based on the Koopman decomposition method [17]. The Asian

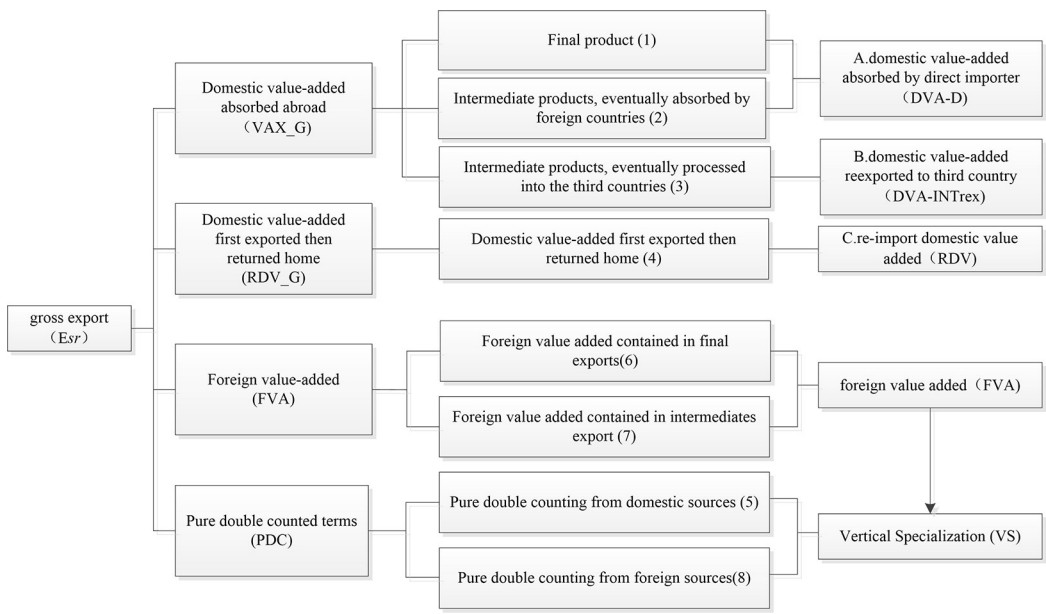

**Fig 1. Complete decomposition of added value (drawn by author according to KWW2014).**

Development Bank (ADB) focused on the main countries in Asia, constructing a time-continuous table of IOT for 61 countries, including southeast Asian countries [18].

## 2.2 Trade facilitation and its impact on international trade

There are some researches on the value-added in trade of some developed countries. However, there is still a lack of research on the trade in value-added of most developing countries, especially those with small trade volume. Existing research on the influencing factors of trade in value-added are mainly focused on export scale [19], FDI [20], R&D input, total factor productivity, production cost, and the size of international trade cost [21]. Although facilitation factors have been included in the scope of trade in value-added influencing factors [22], there is still a gap in the research on the decomposition of trade value and the exploration of facilitation factors to analyse the driving mechanism of different types of export in value-added.

Trade facilitation mainly refers to a country's tariff system and other measures to reduce trade costs in a narrow sense, while it includes trade-related infrastructure and business

**Table 1. Indicators of domestic value-added in trade and their connotation.**

| indicators | connotation and representation |
| --- | --- |
| A. direct domestic value (*DVA-D*) | Domestic added value exports final products to the importing country or is directly absorbed by the importing country. As it reflects the consumption of final products, the value chain is relatively short and reflects the rigid demand of the importing country. |
| B. indirect domestic value (*DVA-INTrex*) | Intermediate products exported by the exporting country to the importing country are transferred to the third country for final consumption by the importing country, which is the main representation of processing trade and reflects the value-added process of the value chain to some extent. |
| C. re-import domestic value (*RDV*) | It reflects the participation mode of the returned value chain and the strengthening of the exporter's control over both ends of the "smile curve". It is the main representation of offshore outsourcing and the embodiment of the new situation of foreign trade. |

environment in a broad sense [23, 24]. At present, most international organizations mainly refer trade facilitation to simplify and coordinate the relevant procedures involved in international investment and international trade activities to solve trade problems caused by conflicts of interest, institutional constraints, and lack of knowledge to reduce trade costs. With trade facilitation, countries can improve the efficiency of transporting commodities[25, 26], and expand the volume of trade [27, 28].

Most of the indicators for measuring trade facilitation in academic studies were based on the framework proposed by the World Bank, including market access & tariffs and trade barriers, customs efficiency, ICT, and infrastructure construction [23, 24]. In addition, some descriptions define trade facilitation from the perspective of the business environment [25]. Based on existing studies, this study established a conceptual framework of trade facilitation (Fig 2), to seek appropriate trade facilitation variables and to further discuss the driving mechanism of the value-added in trade network.

**2.2.1 Market access, tariffs, and trade barriers (mar).** Tariff cost between countries is known as the core cost of the international trade [29]. Tariff reduction was considered as the important symbol of effective economic and trade cooperation between countries, which could open the domestic market effectively, promote market access, integrate into the global trade network, and reduce the cost of trade process effectively [23, 24]. According to existing literature, tariff barriers and market access are the important factors that determine the diversity of regional trade volume and trade products [30–32]. Among them, tariff barriers and market access often have a strong influence in developed countries and regions, such as in southern Europe and non-EU Mediterranean countries.

Research shows that tariff barriers between countries in the FTA are much lower, but non-tariff barriers based on political and economic cooperation between countries still exist.

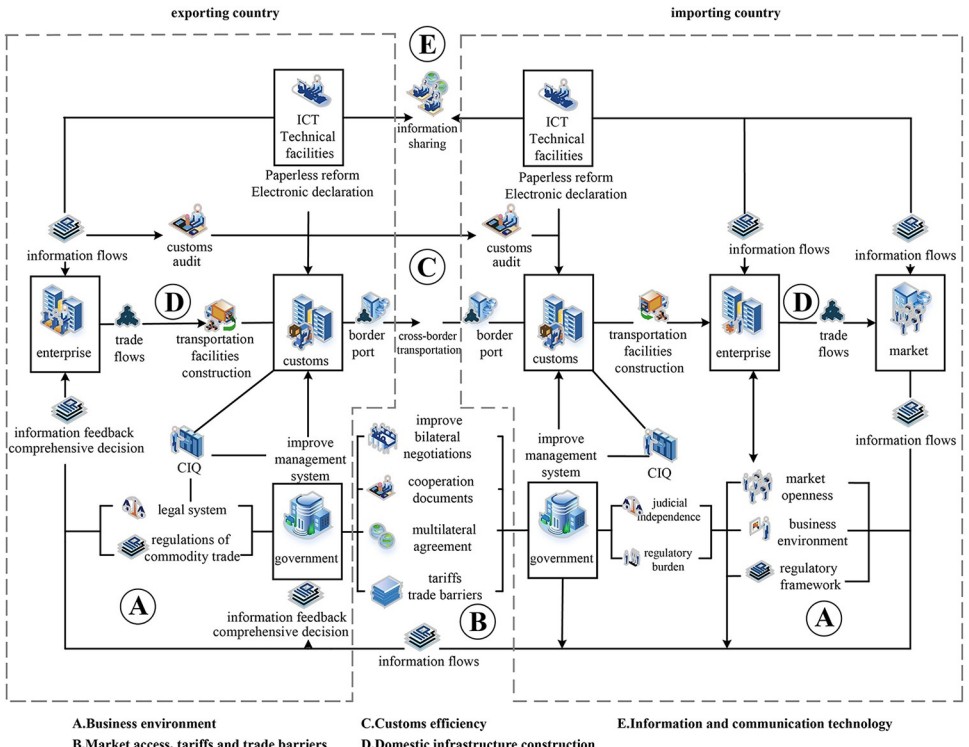

**Fig 2. Connotation of trade facilitation.**

Therefore, *the tariff level* could be regarded as an important indicator of a country's market openness. We selected five indicators from *The Global Enabling Trade Report*, such as *tariff rate*, *complexity of tariffs*, *share of duty-free imports*, *tariffs faced*, and *margin of pref. in destination markets*, and calculated their average value to represent the market access degree of a country.

**2.2.2 Customs efficiency (cus).** Customs efficiency proved to be another trade facilitation factor affecting trade flows and the types of commodities exported [33–35]. It is argued that the impact of customs efficiency on the international trade volume are mainly about the increase in export time and cost delay, due to the large number of clearance documents [32]. Currently, studies on trade facilitation mainly focus on Africa and the Middle East [31], where most countries have the most complex customs procedures and documents in the customs clearance process [36]. Some studies also argued that the high cost of customs will lead to increased uncertainty among traders and generate significant costs and delays [30]. As a result, the improvement of the customs environment of the importing countries has proved to bring far higher benefits in terms of trade flow than other trade facilitation factors.

To comprehensively consider the development degree of the national customs environment, we apply the average value of the eight indicators from *The Global Enabling Trade Report* to evaluate the customs efficiency based on the *clearance efficiency* and the *clearance time and cost*.

**2.2.3 Information and communication technology(ict).** Compared with other factors, the development of ICT has an indirect effect on international trade. However, its effect has been particularly evident since 2000 [37]. Based on paperless and electronic ICT, customs clearance reform greatly shortened the time of acceptance of the goods and reduced the cost [20, 38]. In the information era, the use of automated programs, files, and electronic exchange becomes increasingly important in the application of the risk management programs [35]. In addition, trade facilitation represented by e-commerce plays an important role in international trade. Therefore, we select *averages of internet users*, *fixed-broadband internet subscriptions*, and the *government online service index* to reflect a country's ICT development level.

**2.2.4 Infrastructure construction (inf).** Improved infrastructure has proven to be important in increasing regional trade and diversifying exports in the Trans-Mekong Region of Southeast Asia. The improvement of transport infrastructure is conducive to the reduction in transport costs, thus generating economic externalities. Stone and Strutt [39] argued that the synergy effect of enterprise development along the economic corridor in southeast Asia drives the spillover effect of the regional economy, promotes the extension of the production network, and further improves trade facilitation. Marti, Puertas, and García [31] revealed that the efficiency of the transport of goods has also been proved to be one of the important factors restricting trade flows in the Middle East. The more complex the types of goods and commodities exported, the more transport infrastructure was impacted. From the aspects of infrastructure construction quality and logistics capacity, we apply the average values of thirteen indicators from *The Global Enabling Trade Report* to analyze the impacts of transportation infrastructure construction on trade cooperation.

**2.2.5 Business environment (bus).** The domestic trade environment, including the business environment and the government environment, plays indirect but important roles in a country's foreign trade. Improvements in the domestic regulatory environment have proven to be beneficial to the volume of trade flows from South Africa's exports to other African countries in southern and eastern Africa [33]. The case of southern Europe also showed that the domestic business environment, including the formulation of procedures or documents related to trade laws and regulations and the appeal procedure, not only determined the efficiency of customs, but also considered as an important criterion for foreign suppliers to choose

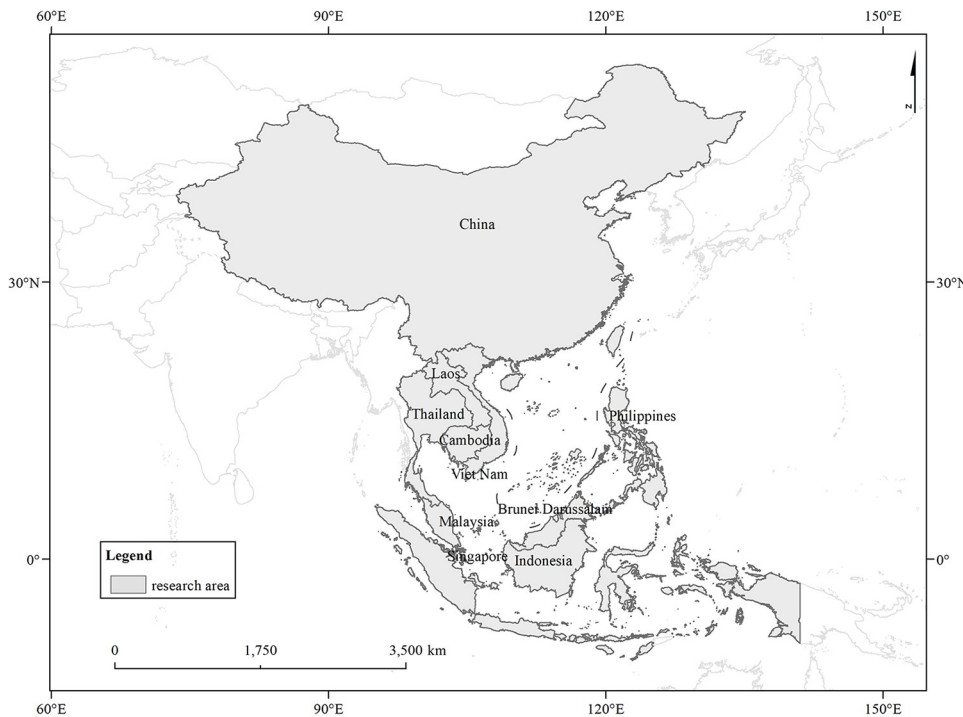

**Fig 3. Research area.**

export products. Moreover, the country's trade environment would affect its cooperation with foreign countries [34, 35]. Therefore, five indicators, such as *protection of the property*, *efficiency and accountability of public institutions*, *access to finance*, *openness to foreign participation*, and *physical security*, were expected to influence the added value trade networks.

## 3. Methodology and data

### 3.1 Research area

The study area includes China and nine members of ASEAN, as Malaysia, Indonesia, Thailand, Philippines, Singapore, Brunei, Vietnam, Laos, and Cambodia. As the original data of the decomposition of export value-added in this study comes from the ADB's Multi-Regional Input-Output Table, Myanmar is not included due to limited data sources and statistics. In addition, due to the statistical calibre, China in this study only includes mainland China; Hong Kong, Macao, and Taiwan were not included (Fig 3).

### 3.2 Methodology

**3.2.1 Decomposition of gross trade flows at bilateral-sector level.** Based on Wang's calculation method [14, 15, 40], we decomposed total exports into different added value components and double-counting items, to achieve the complete decomposition of total exports and build a bridge between official trade and total trade bilateral-sector level. And the research of trade flows between countries and sectors of the world is performed using a multi-regional input-output table (Table 2).

In Table 2, $X$ is the output matrix, $A$ is the input coefficient matrix, $I$ is the identity matrix, $B$ is the inverse matrix of Leontief, and $Y$ is the final product matrix. In the model with $G$ countries and $N$ departments, there will be: $AX+Y = X$ and $X = (I–A)^{-1}Y = BY$. After expansion, the

**Table 2. Multi-regional input-output table.**

| | | Intermediate Use | | | | | Final Demand | | | Total |
|---|---|---|---|---|---|---|---|---|---|---|
| | | Country 1 | | | ... | Country G | Country1 | ... | Country G | |
| Intermediate input | Country1 | Sector1 | ... | Sector N | | | $Y^{sr}$ | | | $X^r$ |
| | | ... | | | | | | | | |
| | | Sector N | | | | | | | | |
| | ... | $X^{sr}$ | | | | | | | | |
| | Country G | | | | | | | | | |
| Value added | | $V^s$ | | | | | | | | |
| Total input | | $X^s$ | | | | | | | | |

decomposition formula (eight parts) for exporting from Country $s$ to Country $r$ can be obtained.

$V^s$ (1×N) is defined as the direct value added coefficient vector, $VB$ is the complete added value coefficient matrix, and $\mu$ (N×1) is the column vector with all 1s. According to the backward inter-industry relations, the final products of Country s are decomposed according to the direction of the value source, that is:

$$\sum_{r \neq s}^{G} V^r B^{rs} + V^s B^{ss} = \mu, \tag{1}$$

where $Z^{sr}$ is defined as the intermediate product export of Country $s$ to Country $r$, and the domestic Leontief inverse matrix of Country $s$ is set as $L^{ss} = (I-A^{ss})^{-1}$. Similarly, $L^{rr}$ and $L^{tt}$ can be set. Because of

$$E_s = \sum_{r \neq s}^{G} E^{rs} = \sum_{r \neq s}^{G} (A^{sr}X^r + Y^{sr}) = Y^s - A^{ss}X^s - Y^{ss} \tag{2}$$

we can obtain:

$$X^r = L^{rr}Y^{rr} + L^{rr}E^r \tag{3}$$

$$Z^{sr} = A^{sr}X^r = A^{sr}L^{rr}Y^{rr} + A^{sr}L^{rr}E^r \tag{4}$$

Finally, the total exports from Country $s$ to Country $E^{sr}$ are obtained:

$$E^{sr} = (V^s B^{ss})^T \# Y^{sr} ① + (V^s B^{ss})^T \# (A^{sr} B^{rr} Y^{rr}) ②$$

$$+(V^s B^{ss})^T \# (A^{sr} \sum_{t \neq s,r}^{G} B^{rt} Y^{tt} + A^{sr} B^{rr} \sum_{t \neq s,r}^{G} Y^{rt} + A^{sr} \sum_{t \neq s,r}^{G} B^{rt} \sum_{u \neq s,r}^{G} Y^{tu}) ③$$

$$+(V^s B^{ss})^T (A^{sr} B^{rr} Y^{rs} + A^{sr} \sum_{t \neq s,r}^{G} B^{rt} Y^{ts} + A^{sr} B^{rs} Y^{ss}) ④$$

$$+[(V^s B^{ss})^T \# (A^{sr} B^{rr} \sum_{t \neq s}^{G} Y^{rs}) + (V^s B^{ss} \sum_{t \neq s}^{G} A^{st} B^{ts})^T \# (A^{sr} X^r)] ⑤$$

$$+[(V^r B^{rs})^T \# Y^{sr} + (\sum_{t \neq s,r}^{G} V^t B^{ts})^T \# Y^{sr})] ⑥$$

$$+[(V^r B^{rs})^T \# (A^{sr} L^{rr} Y^{rr} + (\sum_{t \neq s,r}^{G} V^t B^{ts})^T \# A^{sr} L^{rr} Y^{rr})] ⑦$$

$$+[(V^r B^{rs})^T \# (A^{sr} L^{rr} E^{r*}) + (\sum_{t=s,r}^{G} V^t B^{ts})^T \# A^{sr} L^{rr} E^{r*})] ⑧ \quad (5)$$

where # is the dot product of the partitioned matrix, the superscript T is the transpose matrix, and ① − ⑧ corresponds to eight items in Fig 1. The calculation method and code in MATLAB are used according to the existing study [41]. The details are omitted due to the limitation of the length of the article.

**3.2.2 Social network analysis.** Social Network Analysis (SNA) mainly focuses on "Ties" data and studies the structural characteristics of networks from a quantitative perspective. The degree of nodes' core in the network is identified and measured by the degree of centrality. Centrality consists of three forms: degree, betweenness, and closeness centrality. Considering that all kinds of value-added in trade networks were full networks and had eliminated orientation, degree centrality was adopted. The formula is:

$$C_D(n_i) = \sum_{j}^{n-1} X_{ji} \quad (6)$$

where $C_D(n_i)$ is the degree centre degree, and $X_{ji}$ is the relationship between nodes. In this study, the calculation formula is:

$$C_D(n_i) = \sum_{j}^{n-1} V_{ij} + \sum_{j}^{n-1} V_{ji} \quad (7)$$

where $C_D(n_i)$ is the degree centrality of a country $i$, $V_{ij}$ is the added value of trade from country $i$ to country $j$, and $V_{ji}$ is the added value of trade from country $j$ to country $i$. The degree centrality of country $i$ is also the value-added in trade link intensity of the research region. Normally, the higher the $C_D(n_i)$, the higher the country's position in the value-added in trade network.

**3.2.3 Trade facilitation index and QAP analysis.**

*(1) Trade facilitation index.* Based on the existing studies, this study applies 33 indicators from five pillars including domestic and foreign market access (*mar*), customs efficiency (*cus*), infrastructure construction (*inf*), information and communication technology (*ict*), and business environment (*bus*), to construct the evaluation index system of trade facilitation [23, 24]. And the data of the indicators were derived from *The Global Enabling Trade Report* (Table 3).

As 33 indicators have different value ranges and action directions, we standardised each indicator to turn it into a benefit indicator whose value range is [0,7]. The larger the index, the stronger the degree of facilitation reflected by this index. The value of each pillar is the average

**Table 3. The trade facilitation evaluation index system.**

| Pillar | The indicators | interval | direction |
|---|---|---|---|
| Domestic and foreign market access (*mar*) | Tariff rate% | [0,100] | - |
| | Complexity of tariffs | [0,100] | + |
| | Share of duty-free imports% | [0,100] | + |
| | Tariffs faced% | [0,100] | - |
| | Margin of pref. in destination markets | [0,100] | + |
| Customs efficiency (*cus*) | Customs services index | [0,1] | + |
| | Efficiency of the clearance process | [1,5] | + |
| | Time to import (h) | Based on actual value | - |
| | Cost to import (USD) | Based on actual value | - |
| | Time to export (h) | Based on actual value | - |
| | Cost to export (USD) | Based on actual value | - |
| | Irregular payments and bribes: imports/exports | [1,7] | + |
| | Customs transparency index | [0,1] | + |
| Infrastructure construction (*inf*) | Available airline seat kilometres | Based on actual value | + |
| | Quality of air transport infrastructure | [0,100] | + |
| | Quality of railroad infrastructure | [1,7] | + |
| | Liner Shipping Connectivity Index | [0,100] | + |
| | Quality of port infrastructure | [1,7] | + |
| | Road quality index | [1,7] | + |
| | Paved roads % | [0,100] | + |
| | Ease and affordability of shipment | [1,5] | + |
| | Logistics competence | [1,5] | + |
| | Tracking and tracing ability | [1,5] | + |
| | Timeliness of shipments to destination | [1,5] | + |
| | Efficiency of transport mode change | [1,5] | + |
| | Postal service efficiency | [1,7] | + |
| Information and communication technology (*ict*) | Internet users % | [0,100] | + |
| | Fixed-broadband Internet subscriptions | [0,100] | + |
| | Government Online Service Index | [0,1] | + |
| Business environment (*bus*) | Protection of property | [1,7] | + |
| | Efficiency and accountability of public institutions | [1,7] | + |
| | Access to finance | [1,7] | + |
| | Openness to foreign participation | [1,7] | + |
| | Physical security | [1,7] | + |

value of the indicator it contains. The trade facilitation index (TFI) is the sum of the five pillars.

*(2) Quadratic Assignment analysis.* Quadratic Assignment analysis(QAP) is based on resampling, and the correlation coefficient between two matrices is given by comparing the similarity of each lattice value in two 1-mode N × N matrices. This study applies QAP to identify the impacts of trade facilitation factors on value-added in trade.

In QAP, one network is an observed network matrix $C_{(n \times n)}$ while the other is a model or expected network matrix $P_{(n \times n)}$. The algorithm proceeds in two steps. In the first step, it computes Pearson's correlation coefficient $R^2$ (plus simple matching, Jaccard, Goodman, Kruskal, Gamma, and Hamming distance) between corresponding cells of the two data

matrices:

$$C = \begin{bmatrix} 1 & C_{11} & \ldots & C_{1,n} \\ 1 & C_{21} & \ldots & C_{2,n} \\ \ldots & \ldots & \ldots & C_{3,n} \\ 1 & C_{n1} & \ldots & C_{n,n} \end{bmatrix} \qquad P = \begin{bmatrix} 1 & P_{11} & \ldots & P_{1,n} \\ 1 & P_{21} & \ldots & P_{2,n} \\ \ldots & \ldots & \ldots & P_{3,n} \\ 1 & P_{n1} & \ldots & P_{n,n} \end{bmatrix} \qquad (8)$$

$$R^2 = [\text{cov}(C,P)/\text{sqrt}(D(C)*D(P))]^2 \qquad (9)$$

In the second step, it randomly permutes rows and columns (synchronously) of one matrix (the observed matrix, if the distinction is relevant) and recomputes the correlation and other measures. The second step is performed hundreds of times to compute the proportion of times that a random measure is larger than or equal to the observed measure calculated in step 1. A low proportion ($< 0.05$) suggests a strong correlation between the matrices that is unlikely to have occurred by chance [42].

First, through QAP regression analysis, the significance and strength between trade facilitation and traditional influencing factors in the value-added in trade network were compared. The econometric model constructed in this study is as follows:

$$\ln(manu_{ijt}) = f[\ln(sum{-}FTA_{ijt}), \ln(controls_{ijt})] \qquad (10)$$

where $i$, $j$, and $t$ represent export country, import country, and year. The explained variable *manu* represented the *DVA-D* value, *DVA-INTrex* value, or *RDV* value from the sum of export country $i$ and import country $j$ in the corresponding regression equation. *Sum-FTA$_{ijt}$*, *sum-cos$_{ijt}$*, *sum-inf$_{ijt}$*, *sum-ict$_{ijt}$*, and *sum-bus$_{ijt}$* represent the trade facilitation value from country $i$ to country $j$ and *controls$_{ijt}$* was the control variable. In this study, two classic variables in international trade were selected, namely, the size of the bilateral economy (*sum-gdp$_{ijt}$*) and the geographical distance between the two countries (*diff-geo$_{ijt}$*).

The matrix of TFI ($C_{sum\text{-}TFI}$) and its five pillars ($C_{sum\text{-}mar}$, $C_{sum\text{-}cos}$, $C_{sum\text{-}inf}$, $C_{sum\text{-}ict}$, $C_{sum\text{-}bus}$) in this study were evaluated by adding the value of trade facilitation in import country $i$ and export country $j$. As the realisation of value-added export involved crossing border ports at least twice and was affected by the business environment of importing and exporting countries. Therefore, it was not only subject to the degree of domestic trade facilitation, but also closely related to the trade facilitation of export markets. The matrix of *GDP* ($C_{sum\text{-}gdp}$) and was evaluated by adding the *GDP* of export country $i$ and import country $j$ (Table 4).

Second, through QAP regression, we explore the driving mechanism of facilitation factors on the value-added in trade network of CAFTA. According to the regression results, the development level of trade facilitation was further subdivided as follows:

$$\ln(manu_{ijt}) = f[\ln(sum{-}bus_{ijt}), \ln(sum{-}mar_{ijt}), \ln(sum{-}cos_{ijt}), \ln(sum{-}inf_{ijt}), \ln(sum{-}ict_{ijt})] \quad (11)$$

Through QAP correlation and regression analysis, the corresponding factors to the value-added in trade network was clarified further. And the value-added in trade scale, economic scale, and geographical distance variables were logarithmic to keep them in line with the order of magnitude of trade facilitation variables to reduce heteroscedasticity.

## 3.3 Data resources

The data of value-added in trade were based on the multi-regional input-output table by the ADB, which has been continuously compiled since 2010 and includes 61 countries and 35

**Table 4. Independent and dependent variables in QAP analysis.**

|  | variables | Instructions |
|---|---|---|
| dependent variables | $DVA\text{-}D_{ijt}$ | The sum of the value of $DVA\text{-}D$ between country $i$ and country $j$ in $t$ year. |
|  | $DVA\text{-}INTrex_{ijt}$ | The sum of the value of $DVA\text{-}INTrex$ between country $i$ and country $j$ in $t$ year. |
|  | $RDV_{ijt}$ | The sum of the value of $RDV$ between country $i$ and country $j$ in $t$ year. |
| independent variables | $sum\text{-}FTA_{ijt}$ | The sum of the value of $FTA$ between country $i$ and country $j$ in $t$ year. |
|  | $sum\text{-}mar_{ijt}$ | The sum of the value of $mar$ between country $i$ and country $j$ in $t$ year. |
|  | $sum\text{-}cos_{ijt}$ | The sum of the value of $cos$ between country $i$ and country $j$ in $t$ year. |
|  | $sum\text{-}inf_{ijt}$ | The sum of the value of $inf$ between country $i$ and country $j$ in $t$ year. |
|  | $sum\text{-}ict_{ijt}$ | The sum of the value of $ict$ between country $i$ and country $j$ in $t$ year. |
|  | $sum\text{-}bus_{ijt}$ | The sum of the value of $bus$ between country $i$ and country $j$ in $t$ year. |
|  | $sum\text{-}gdp_{ijt}$ | The sum of the value of $GDP$ between country $i$ and country $j$ in $t$ year. |
|  | $diff\text{-}geo_{ijt}$ | The distance between country $i$ and country $j$ in $t$ year. |

sectors. The methods and codes of decomposition of gross trade flows at the bilateral-sector level came from the database compiled by the research institute for GVC of the University of International Business and Economics.

Trade facilitation data came from *The Global Enabling Trade Report* released by The World Economy Forum. Since *The Global Enabling Trade Report* was officially released in 2008, we could only analyzed the impact factors from 2008. The country's GDP from the World Bank was adjusted to 2010. The geographical distance between countries came from the Center for Perspective Research and International Information Technology (CEPII).

## 4. Results and analysis

### 4.1 Decomposition of gross trade flows

**4.1.1 Differences of gross export in CAFTA.** According to the calculation of the ADB data, from 2001 to 2016, within the development of CAFTA, its exports increased significantly. As for countries, the highest growth country was Laos, which exported to other CAFTA countries increasing from 480 million USD in 2001 to 5,954 million USD in 2016, with an average annual growth rate of 18.27%. At the same time, Vietnam, Cambodia, the Philippines, and Indonesia all grew by more than 14%. Singapore, Brunei, and Malaysia had slower export growth of less than 10% (Fig 4). China's exports to other CAFTA countries increased from 18.18 billion USD in 2001 to 138.05 billion USD in 2016, with an average annual growth rate of 14.47%, ranking sixth among the ten countries. Based on the decomposition of gross trade, the growth rate of domestic added value of each country's export was higher than that of total export. Moreover, compared with other value components of exports, the growth rate of *RDV* in each country was the highest. As for countries, the growth rate of *RDV* of China's exports to other CAFTA countries was 50.74%, only lower than Laos.

The proportion of each component of the decomposition of export value was further analyzed (Fig 5). All countries' *DVA* proportion increased from 2001 to 2016. In 2001, Vietnam's *DVA* proportion was the highest, at 60.38%, and other countries *DVA* proportions were less than 60%, although Thailand and Malaysia were relatively high. China's *DVA* proportion accounted for 48.27%, only higher than that of Laos. In 2016, the *DVA* proportion of all countries further increased, with Vietnam and Brunei accounting for more than 70%. China's *DVA* proportion increased only slightly, to 55.55%. From 2001 to 2016, the *DVA* proportions that have increased significantly, to some extent embodied the advancement of CAFTA trade cooperation, industry

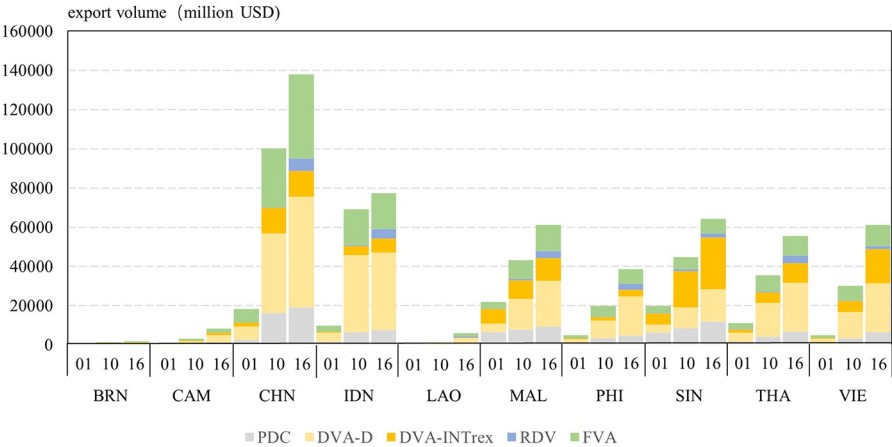

**Fig 4. Exports and the value of different component between China and ASEAN countries in 2001, 2010, and 2016.**

upgrades, and the product of trade structure optimisation. This was the result of the use of domestic raw materials, and local optimisation and service trade enterprises increased earnings.

The analysis of different component proportions exported by countries is given in Table 5. From 2001 to 2016, the proportion of *DVA-INTrex* in total exports increased, while the proportion of *FVA* decreased. In general, in the production and trade network of CAFTA, all countries' status in the RVC continuously improved. In terms of countries, *DVA-INTrex* of Singapore was always higher than that of *FVA*, which means it had a relatively high position in the production and trade network of CAFTA. It could be partly explained by that Singapore mainly produces intermediate commodities for trading countries, which is in the upstream link of the value chain. In 2001, the proportion of *DVA-INTrex* in total exports of Vietnam and Thailand was significantly lower than that of *FVA*; while in 2016, it was significantly higher than that of *FVA*. This reflects the improvement in the production division in regional trade of Vietnam and Thailand. However, the opposite is true in Malaysia. In 2001, *DVA-IN-Trex* accounted for 10.19% of China's total exports, while *FVA* accounted for 37.91%; in 2016, *DVA-INTrex* accounted for 9.39%, while *FVA* accounted for 31.11%. The results show that China had relatively low position in the CAFTA RVC before 2016.

Furthermore, the proportion of *RDV* in each country increased significantly. In 2001, the *RDV* share of the CAFTA regional trade network was the highest in Malaysia (0.90%), while

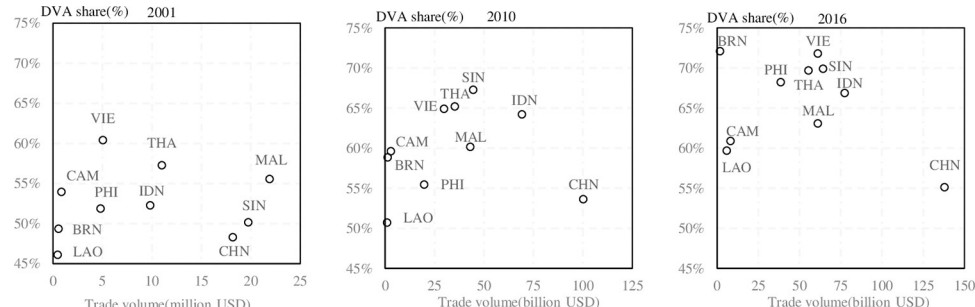

**Fig 5. DVA share of mutual export between China and ASEAN countries in different years.**

**Table 5. Total decomposition of export value of China and ASEAN countries in different years.**

| year | components | BRN | CAM | CHN | IDN | LAO | MAL | PHI | SIN | THA | VIE |
|------|-----------|-----|-----|-----|-----|-----|-----|-----|-----|-----|-----|
| 2001 | DVA-D | 45.64% | 45.95% | 38.00% | 47.73% | 42.75% | 20.70% | 42.46% | 20.36% | 40.83% | 49.22% |
| | DVA-INTrex | 3.61% | 7.91% | 10.19% | 4.10% | 3.20% | 33.94% | 9.20% | 28.95% | 16.00% | 10.79% |
| | RDV | 0.09% | 0.06% | 0.08% | 0.41% | 0.13% | 0.90% | 0.19% | 0.82% | 0.41% | 0.37% |
| | FVA | 31.82% | 29.12% | 37.91% | 31.10% | 30.45% | 15.28% | 30.86% | 18.13% | 28.15% | 28.99% |
| | PDC | 18.84% | 16.95% | 13.82% | 16.65% | 23.47% | 29.18% | 17.29% | 31.74% | 14.60% | 10.63% |
| 2010 | DVA-D | 49.34% | 45.90% | 40.25% | 56.97% | 44.75% | 36.72% | 45.96% | 23.54% | 48.24% | 45.03% |
| | DVA-INTrex | 9.19% | 13.52% | 13.24% | 6.77% | 5.67% | 21.99% | 8.93% | 41.20% | 15.66% | 18.40% |
| | RDV | 0.31% | 0.19% | 0.12% | 0.45% | 0.27% | 1.44% | 0.56% | 2.50% | 1.28% | 1.45% |
| | FVA | 29.26% | 25.42% | 30.11% | 26.62% | 36.65% | 22.03% | 27.73% | 13.44% | 22.77% | 24.05% |
| | PDC | 11.91% | 14.97% | 16.29% | 9.20% | 12.66% | 17.83% | 16.83% | 19.32% | 12.05% | 11.07% |
| 2016 | DVA-D | 54.55% | 44.38% | 40.99% | 51.29% | 41.49% | 38.41% | 51.84% | 25.35% | 44.65% | 40.54% |
| | DVA-INTrex | 9.83% | 16.15% | 9.39% | 9.02% | 4.75% | 18.76% | 8.41% | 41.09% | 18.12% | 28.58% |
| | RDV | 7.68% | 0.35% | 4.72% | 6.53% | 13.44% | 5.90% | 7.97% | 3.44% | 6.90% | 2.67% |
| | FVA | 17.88% | 23.63% | 31.11% | 23.48% | 24.66% | 21.88% | 19.53% | 11.63% | 18.02% | 17.63% |
| | PDC | 10.06% | 15.50% | 13.80% | 9.67% | 15.67% | 15.06% | 12.25% | 18.49% | 12.30% | 10.59% |

that of Singapore was relatively high (0.80%). In 2010, the highest *RDV* proportion was Singapore (2.51%), and more than 1% in Thailand, Malaysia, and Vietnam. In 2016, the proportion of *RDV* in each country continued to increase, especially Laos and Thailand. The transformation indicates that raw materials or intermediate goods were produced in the country, exported to other countries for processing and manufacturing, and then returned to the country through import from the export market or third country to meet the domestic production and consumption demand. This mode of inward value chain participation, which is first in and then out, was deepening, and the international step length of production between enterprises was extended. Furthermore, the results reveal that under the framework of CAFTA, especially since 2010, the production and trade links between CAFTA countries have gradually moved to the upstream high value-added links, and the RVC system with domestic demand has been constructed.

**4.1.2 Network structure of value-added in trade in CAFTA.** Fig 6 shows the SNA results of three value-added network structures in CAFTA, namely *DVA-D*, *DVA-INTrex*, and *RDV*. First, China's status in the trade network of CAFTA has significantly improved. In 2001, China's *DVA-D* trade network ranked first with a degree of centrality of 14,422.52. However, its *DVA-INTrex* and *RDV*, two trade networks that better reflect the international division of labour and production chain status, ranked only third, with degrees of centrality being 5707.05 and 212.25, respectively. In 2016, China ranked first in degree centrality for *DVA-D*, *DVA-INTrex*, and *RDV*. Second, Singapore and Malaysia's positions in the trade network declined slightly, but overall, they still played a key role in the CAFTA trade network. Singapore's degree of centrality rankings in the *DVA-INTrex* and *RDV* networks were first in 2001, and has moved to a second and fourth respectively in 2016. Malaysia's degree centrality rankings of *DVA-INTrex* and *RDV* networks were both second in 2001, but by 2016 it had moved to third in both cases. Third, Thailand has become the most benefited country after the establishment of CAFTA, with its centrality in the three networks significantly improved. Fourth, Brunei, Cambodia, Laos, and the Philippines had low degrees of centrality and the change was not obvious. And their series effect on the network was relatively low (Table 6).

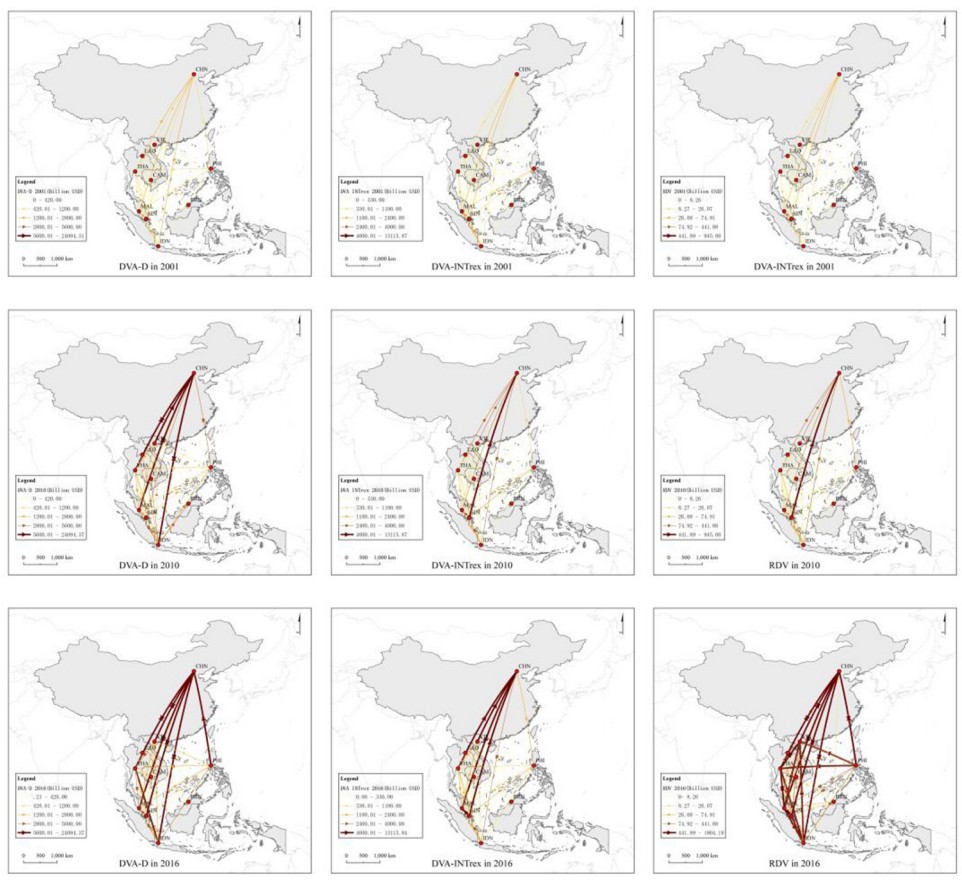

**Fig 6. Value-added in trade network in CAFTA (2001, 2010, 2016).**

## 4.2 The main drivers of value-added in trade network in CAFTA

The QAP results show that the impact of traditional variables on three value-added trade networks is consistent with existing literatures(Table 7). On the one hand, the impacts of total economic scale(*sum-gdp*) are significant and positive to the three value-added in trade

**Table 6. Results of degree centrality of value-added in trade network in CAFTA (2001,2016).**

| country | 2001 | | | | | | 2016 | | | | | |
|---|---|---|---|---|---|---|---|---|---|---|---|---|
| | *DVA-D* | | *DVA-INTrex* | | *RDV* | | *DVA-D* | | *DVA-INTrex* | | *RDV* | |
| | $C_d (n_i)$ | rank | $C_d(n_i)$ | rank | $C_d (n_i)$ | rank | $C_d (n_i)$ | rank | $C_d (n_i)$ | rank | $C_d (n_i)$ | rank |
| BRN | 535.68 | 8 | 187.28 | 8 | 0.55 | 10 | 955.09 | 9 | 478.36 | 10 | 163.14 | 9 |
| CAM | 461.07 | 9 | 93.88 | 9 | 0.56 | 9 | 1613.62 | 8 | 1682.03 | 8 | 114.59 | 10 |
| CHN | 14422.52 | 1 | 5707.05 | 3 | 212.25 | 3 | 84894.92 | 1 | 53975.87 | 1 | 12652.40 | 1 |
| IDN | 10036.16 | 2 | 3813.20 | 4 | 91.11 | 4 | 36500.95 | 2 | 19248.15 | 6 | 6712.38 | 5 |
| LAO | 303.90 | 10 | 45.99 | 10 | 0.67 | 8 | 696.76 | 10 | 763.26 | 9 | 819.34 | 8 |
| MAL | 9699.02 | 3 | 10718.38 | 1 | 283.58 | 1 | 32950.46 | 4 | 24149.80 | 3 | 7621.51 | 3 |
| PHI | 3415.24 | 7 | 2486.64 | 6 | 21.55 | 7 | 14667.43 | 7 | 6965.35 | 7 | 4292.78 | 7 |
| SIN | 7904.27 | 5 | 8909.64 | 2 | 261.23 | 2 | 20255.26 | 5 | 32443.36 | 2 | 7500.00 | 4 |
| THA | 8961.70 | 4 | 3533.18 | 5 | 81.45 | 5 | 34575.24 | 3 | 20031.08 | 5 | 7921.12 | 2 |
| VIE | 4376.51 | 6 | 1039.23 | 7 | 24.14 | 6 | 19483.97 | 6 | 21159.92 | 4 | 6029.31 | 6 |

**Table 7. Results of QAP regression in value-added in trade networks with TFA.**

| year | 2010 | | | 2017 | | |
|---|---|---|---|---|---|---|
| variable | DVA-D | DVA-INTrex | RDV | DVA-D | DVA-INTrex | RDV |
| intercept | -0.609*** | -5.095*** | -5.758*** | -1.544*** | -1.867*** | -4.198*** |
| ln(sum-gdp) | 1.471*** | 1.305*** | 0.691*** | 1.336*** | 1.168** | 0.954*** |
| ln(diff-geo) | -0.857* | -0.692 | -0.145 | -1.051** | -0.986** | -0.622 |
| sum-TFA | 0.063 | **0.144*** | **0.092*** | 0.058 | **0.086*** | **0.146*** |
| $R^2$ | 0.560 | 0.539 | 0.408 | 0.572 | 0.538 | 0.428 |
| adjusted $R^2$ | 0.549 | 0.528 | 0.394 | 0.563 | 0.522 | 0.415 |

*Significant at 0.10

** Significant at 0.05

*** Significant at 0.01.

networks. In 2010, the regression coefficients for *DVA-D*, *DVA-INTrex*, and *RDV* networks were 1.471, 1.305, and 0.691, respectively. In 2016, its regression coefficients for *DVA-D*, *DVA-INTrex*, and *RDV* networks were 1.336, 1.168, and 0.954, respectively. As a result, the size of a country's economy is still a decisive factor in determining its foreign trade. On the other hand, the coefficient of geographical distance between countries is negative, which reveals that the value-added in trade between CAFTA countries is still limited by spatial distance(*diff-geo*). However, it is noteworthy that the impact of geographical distance is only significant in the *DVA-D* network in 2010 and the *DVA-D* network and *DVA-INTrex* network in 2016. This could be partly explained by that in the *DVA-D* trade network, the added value of national exports is directly absorbed by importing countries, and the added value is attached to primary products such as agricultural products or bulk commodities that are not easy to transport. Therefore, the influence of geographical distance between countries is significantly negative. For the *DVA-INTrex* and *RDV* trade networks, value-added in trade flows are more dependent on the comparative advantages of countries in regional production networks. As a result, the restriction effect of geographical distance on value-added in trade has been weakened under the progress of globalisation.

The QAP results also show that trade facilitation has a significant positive effect on the *DVA-INTrex* and *RDV* trade networks, which could reflect the intermediate goods trade process, compared to the *DVA-D* network. In 2010 and 2016, the coefficients of trade facilitation for the *DVA-INTrex* network were 0.144 and 0.086, and for the *RDV* network were 0.092 and 0.146. A country's level of trade facilitation directly determines the cost advantage of participating in the international division of labour. If a country lacks such cost advantage in system, infrastructure, administrative burden, business cost, and other aspects, it will be excluded from the GPN [43]. In other words, trade facilitation provides a realistic basis for the division of labour in the value chain of regional production network, reduces the cross-border barriers for intermediate goods, promotes the refinement of intra-product division between countries, and increases the frequency of the return of intermediate goods between economies. As a result, trade facilitation could increase the indirect value-added and returned value-added (Table 8).

Based on QAP results, countries could improve the level of trade facilitation to deepen the regional production network and value flow in CAFTA. In order to further identify which trade facilitation factors influence the intermediate goods trade process most, we applied QAP analysis of the different pillars of trade facilitation, as *sum-bus*, *sum-mar*, *sum-cus*, *sum-inf*, and *sum-ict* to *DVA-INTrex* and *RDV* networks. The results are shown in Table 8.(1) The

**Table 8. QAP results of different TFI pillars' impacts on three value-added in trade networks.**

| year | 2010 | | | | 2017 | | | |
|---|---|---|---|---|---|---|---|---|
| variable | DVA-INTrex | | RDV | | DVA-INTrex | | RDV | |
| | QAP correlation | QAP regression | QAP correlation | QAP regression | QAP correlation | QAP regression | QAP correlation | QAP regression |
| sum-bus | 0.660*** | 0.660*** | 0.598** | 0.598** | 0.610*** | 2.957** | 0.605** | 2.961** |
| sum-mar | 0.155 | - | 0.156 | - | 0.249 | - | 0.278 | - |
| sum-cus | 0.332 | - | 0.315 | - | 0.415* | -0.882 | 0.404* | -0.793 |
| sum-inf | 0.273 | - | 0.186 | - | 0.33 | - | 0.303 | - |
| sum-ict | 0.380 | - | 0.299 | - | 0.424* | 0.299 | 0.395* | 0.117 |
| intercept | | -19.538*** | | -10.753*** | | -15.607*** | | -15.859*** |
| $R^2$ | | 0.719 | | 0.525 | | 0.41 | | 0.405 |
| adjusted $R^2$ | | 0.698 | | 0.491 | | 0.397 | | 0.391 |
| sample size | | 90 | | 90 | | 90 | | 90 |

*Significant at 0.10

** Significant at 0.05

*** Significant at 0.01.

domestic business environment(*bus*) is the key facilitation factor, which has a significant impact on the CAFTA value-added in trade network. As shown in Table 8, the business environment was the only significantly correlated factor for the *DVA-INTrex* and *RDV* networks in 2010. In 2016, the *bus* was the most important factors of the *DVA-INTrex* and *RDV* networks, with coefficients of 2.957 and 2.961. As most CAFTA countries are developing countries, their business environment have larger room to promote, in case of technology, economy, and legal aspects. According to the CAFTA framework, many significant measures could improve business environment and boost intra-trade, such as construction of legislation system, expanding financing channels, improving the rationality of capital flow, protecting property security, and improving the efficiency of commercial institutions.

(2) Customs efficiency (*cus*) and information and communication technology (*ict*) are the second important factors. And they have gradually increased their impacts on two value-added in trade networks since 2010. In 2016, the *cus* and *ict* were -0.882 and -0.793, respectively. On the one hand, literature revealed that customs clearance was the most important factor hindering trade flows between countries, especially in developing countries. It is believed that the systematisation of customs procedures and the consistency of customs rules are conducive to the coordination of trade safety, especially to the import and export of intermediate goods. On the other hand, China and ASEAN countries have been strengthening e-commerce cooperation, which greatly improved the efficiency of intermediate goods trade and service trade. At the same time, China and ASEAN countries should improve their Internet infrastructure, develop digital economy, and encourage ICT use [44].

(3) Domestic and foreign market access (*mar*) and availability and quality of transport infrastructure and services (*inf*) have little effect on the value-added in trade network. As shown in Table 8, the coefficients of market access (*mar*) and transport infrastructure (*inf*) are very little, regardless of QAP correlation or QAP regression. As for market access (*mar*), it is partly because of the low tariff level in CAFTA. When CAFTA was proposed in 2001, it already had achieved zero tariff on 90% of traded goods. Moreover, ASEAN, as a regional cooperation organization, basically realised zero tariff on goods traded between countries

in 2015. However, this result could reveal that trade liberalisation had a limited impact on value-added in trade and international division of labour in CAFTA. As for transport infrastructure (*inf*), the reason may be that the gains from improved transport efficiency were due to increased public expenditure caused by further construction of infrastructure, which is a heavy economic burden for most developing countries.

## 5. Conclusions and discussion

### 5.1 Conclusions

As the largest FTA of developing countries in the world, CAFTA has greatly strengthened the intra-trade flow between China and ASEAN. This will help some countries, whose intermediate goods trade accounts for a large proportion, to better participate in the international division of production and cooperation. Based on the ADB regional input-output table and a total of 66180 sets of data, this study calculated and decomposed the value-added in trade between China and nine ASEAN countries especially focused on the *DVA* trade network. Additionally, through the QAP correlation and regression analysis, this study analysed the influence of the trade facilitation factors on the *DVA* trade network of CAFTA.

The decomposition results show that in the CAFTA trade network, the proportion of *DVA* kept increasing from 2001 to 2016. In other words, most countries have optimized their trade structure, by upgrading industries and products. In the CAFTA trade networks, Singapore, Vietnam, and Thailand were relatively high, while China was relatively low. Second, each country's *RDV* proportion has increased significantly from 2001 to 2016, which means the development of the first out lower turn-back type of value chain participation model in CAFTA countries. Since 2010, most countries in CAFTA have gradually moved upstream high value-added positions, and they are building a RVC system with domestic demand as the core. Third, SNA results demonstrate that China and Thailand were the main beneficiaries after the establishment of CAFTA. And the positions of Singapore and Malaysia in the CAFTA trade network have declined slightly, but they still played a key role.

QAP analysis shows that trade facilitation factors have a positive significant effect on the *DVA INTrex* and *RDV* trade networks, while the impact of traditional variables still mattered. The results reveal that trade facilitation could provide a realistic foundation for the division of regional value chain, which could reduce the products' barriers to cross-border trade. At the same time, trade facilitation can increase the products' back and forth frequency between economies, which may also increase the *DVA-INTrex* and *RDV* in the trade process. For each trade facilitation factors, the domestic business environment (*bus*) was the most important factors for the *DVA-INTrex* and *RDV* trade networks; border administration(*cus*) and availability of ICTs (*ict*) are the second important, with gradually improved impacts on *DVA-INTrex* and *RDV* trade networks; market access (*mar*) and transport infrastructure and services (*inf*) have little effects on the value-added in trade network.

### 5.2 Limitations

There are some limitations of this study. First, the research data still needs supplement. ASEAN consists of ten countries, but the study area only includes nine countries, without Myanmar. As the data on added value of trade come from the multi-regional input-output table compiled by the ADB, which includes nine countries of ASEAN. The absence of Myanmar, to some extent, may cause partial deviation in the analysis of value-added in trade network of CAFTA.

Second, there may be some deficiencies in the model. According to existing studies, for countries where processing trade plays an important role in foreign trade, there will be some errors in the general non-competitive input-output model. Currently, there are non-competitive input-output models that reflect processing trade, but only by decomposing the total exports of individual countries. We try to decompose three parts of domestic value-added (DVA), as the direct value-added (*DVA-D*), indirect value-added (*DVA-INTrex*), and re-import value-added (*RDV*). However, further improvement in the decomposition model is still required and deserves further attention.

## 5.3 Policy implication

For most developing countries, the results show that improving the business environment is the most important way to accelerate the deepening of regional trade and production networks. And improving the institutional building of countries' trade facilitation, could actively promote the trade facilitation process, and create an efficient and transparent trade environment, which plays an important role in deepening the RVC. Meanwhile, developing countries should pay more attention on the modern trade management and relevant policies and legal procedures, which could facilitate regional trade network. Moreover, it is also important to improve the efficiency of customs clearance. promote e-commerce and paperless reform, strengthen the individual's Internet skill, eliminate high-cost transactions and serious time delays, and reduce administrative corruption. Furthermore, it will be necessary to maintain trade safety coordination and enhance risk control.

For managers and planners, the results suggest that GIS and econometric analysis combined with statistics can provide useful information for analysis and decision making. For researchers, this study reveals the value flow in the process of China-ASEAN regional trade from the perspective of added value. Unlike traditional analysis, it focuses on the decomposition of value flows in the process of regional bilateral trade, which is not common. Through ArcGIS, the trade flow between countries is better revealed, which makes the trade flow visible and gives people a relatively intuitive feeling. And it also provides a new idea for interdisciplinary research, especially the economic, geographical, and trade research.

## Author Contributions

**Conceptualization:** Zhouying Song.

**Data curation:** Yang Yu, Zhouying Song.

**Formal analysis:** Yang Yu.

**Investigation:** Yang Yu, Zhouying Song.

**Methodology:** Yang Yu.

**Project administration:** Yang Yu.

**Resources:** Zhouying Song.

**Writing – original draft:** Yang Yu, Zhouying Song.

**Writing – review & editing:** Yang Yu.

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
