## [Decision Letter · Decision Letter 0]

18 Apr 2023

PONE-D-23-06237Main drivers of regional value chains in CAFTA: Does trade facilitation matter?PLOS ONE

Dear Dr. Song,

Thank you for submitting your manuscript to PLOS ONE. After careful consideration, we feel that it has merit but does not fully meet PLOS ONE’s publication criteria as it currently stands. Therefore, we invite you to submit a revised version of the manuscript that addresses the points raised during the review process.

Authors need to focus on:1) explaining the abbreviations upfront 2) discussion needs improvement. ==============================

We look forward to receiving your revised manuscript.

Kind regards,

Muhammad Khalid Bashir, PhD

Academic Editor

PLOS ONE

Journal Requirements:

"Priority Research Program of Chinese Academy of Sciences（XDA20010102）

The polishing fee has been paid for this project, and the layout fee can be reim"

5. We note that Figures 3 and 6  in your submission contain [map/satellite] images which may be copyrighted. All PLOS content is published under the Creative Commons Attribution License (CC BY 4.0), which means that the manuscript, images, and Supporting Information files will be freely available online, and any third party is permitted to access, download, copy, distribute, and use these materials in any way, even commercially, with proper attribution. For these reasons, we cannot publish previously copyrighted maps or satellite images created using proprietary data, such as Google software (Google Maps, Street View, and Earth). For more information, see our copyright guidelines: http://journals.plos.org/plosone/s/licenses-and-copyright.

a. You may seek permission from the original copyright holder of Figures 3 and 6  to publish the content specifically under the CC BY 4.0 license.  

Reviewers' comments:

Reviewer's Responses to Questions

**Comments to the Author**

1. Is the manuscript technically sound, and do the data support the conclusions?

Reviewer #1: Yes

Reviewer #2: Yes

2. Has the statistical analysis been performed appropriately and rigorously? 

Reviewer #1: Yes

Reviewer #2: Yes

3. Have the authors made all data underlying the findings in their manuscript fully available?

Reviewer #1: Yes

Reviewer #2: No

4. Is the manuscript presented in an intelligible fashion and written in standard English?

Reviewer #1: Yes

Reviewer #2: Yes

5. Review Comments to the Author

Reviewer #1: The authors worked on interesting research topic "Main drivers of regional value chain in CAFTA: Does trade facilitation matters". The article is acceptable after some minor revisions. The authors used many abbreviations - especially in abstract - some of them needs to be explained right from start of the article. Further, the article lacks the discussion. The section 5 of the article is "Conclusions and Discussion". In the subsections no discussion is found. The authors should discuss the results before the conclusion by adding separate section "discussion" which should be sufficient enough according to the size of the results.

Reviewer #2: This study claims to make the three contributions to the international trade: (1) analysing the regional value chains (RVCs) from the perspective of value-added in trade at the bilateral level, (2) systematic analyses of the influences of facilitation factors on the value-added in trade, (3) spatial analysis (and mapping) to develop a descriptive understanding of the network of value-added in trade in CAFTA to show RVCs.

To further improve the quality of the manuscript, the following suggestions are proposed:

1. In the abstract, also add that for what the abbreviation DVA stands

2. No doubt, the presentation of a diagrammatic conceptual framework is an elaborative contribution of the study. But it would increase the understand about the trade facilitating variables, if each of the different sections (A, B, C, D, and E) of the diagram (Fig.2) are a bit explained further.

3. There is need to add the theoretical underpinnings of the study as well, for example different models of trade, pertinent to this study, like, the gravity theory of international trade etc.

4. Fig.6 (depicting SNA results) is fade and does not clearly exhibit the trade related networking among the CAFTA countries. Similarly, there is no explanation about the figure in terms of what the figure is showing through dark and dim multicolored lines in different time periods.

5. Table 8 claims to exhibit QAP results of different TFI pillars’ impacts on three value-added in trade networks, whereas only the finding of two trade networks (DVA-INTrex and RDV networks) are there.

6. QAP analyses (QAP correlations and QAP regressions for the years 2010 and 2017) are done to investigate the influence of the trade facilitation factors on the intermediate goods trade. Those trade facilitation factors or pillars are: domestic business environment(bus), domestic and foreign market access (mar), customs efficiency (cus), availability and quality of transport infrastructure and services (inf), and information and communication technology (ict). But in the QAP regressions (as evident from the findings given in Table 8), why all the five different pillars of trade facilitation are not included.

7. The findings of the study are discussed least in comparison with the relevant previous literature

8. Policy implications (section 5.3 of the manuscript) could not be so specific, like stated in the lines 574 and 575: “promote e-commerce and paperless reform, strengthen the individual's Internet skill, eliminate high cost transactions and serious time delays, and reduce administrative corruption”.

6. PLOS authors have the option to publish the peer review history of their article (what does this mean?). If published, this will include your full peer review and any attached files.

Reviewer #1: No

Reviewer #2: No

---

## [Author Response · Author response to Decision Letter 0]

4 Jul 2023

Dear editors and reviewers,

Thank you very much for these detailed and useful feedbacks on our paper. We have considered your comments and have incorporated almost all your suggestions in the revised version of our paper. We also have a detailed response to each comment in the below. In the paper, the changes made are highlighted in yellow. We feel that by revising our paper according to your constructive suggestions, the paper has been greatly improved.

Comments from the editors and reviewers:

- Referee 1-

The authors worked on interesting research topic "Main drivers of regional value chain in CAFTA: Does trade facilitation matters". The article is acceptable after some minor revisions. 

First, the authors used many abbreviations - especially in abstract - some of them needs to be explained right from start of the article. 

Response: Thanks a lot for your valuable comments. We have supplemented our previous oversight by explaining abbreviations as China and ASEAN Free Trade Area (CAFTA), domestic value added (DVA), returned added value (RDV), and domestic value added reexported to third country (DVA INTrex) in the abstract.

Further, the article lacks the discussion. The section 5 of the article is "Conclusions and Discussion". In the subsections no discussion is found. The authors should discuss the results before the conclusion by adding separate section "discussion" which should be sufficient enough according to the size of the results. 

Response: Thanks for your valuable comments. According to your and the second reviewer’s suggestion, we tried to add the discussion part of the results. We have added a discussion section-Section 4.3, in which we discussed the results and compared them with relevant previous literature. For example, we compared the research results of trade facilitation on DVA-D, DVA-INTrex, and RDV with previous literature (Line 576-589), and found out that the results showed consistency with previous literature. Moreover, we pointed out that this study further revealed that in regions like CAFTA where tariffs and trade barriers between internal countries are relatively low, the key factor affecting the value of DVA-INTrex and RDV is the business environment(bus). Please see the revised part in Section 4.3 (Line 590-598).

- Referee 2-

To further improve the quality of the manuscript, the following suggestions are proposed:

1. In the abstract, also add that for what the abbreviation DVA stands

Response: Thanks a lot for your valuable comments. We have supplemented our previous oversight by explaining abbreviations as China and ASEAN Free Trade Area (CAFTA), domestic value added (DVA), returned added value (RDV), and domestic value added reexported to third country (DVA INTrex) in the abstract.

2. No doubt, the presentation of a diagrammatic conceptual framework is an elaborative contribution of the study. But it would increase the understand about the trade facilitating variables, if each of the different sections (A, B, C, D, and E) of the diagram (Fig.2) are a bit explained further.

Response: Thanks for your positive and valuable comments. We are very sorry for the confusing signs, as sections (A, B, C, D, and E) of the diagram (Fig.2) were not in a one-to-one correspondence to the explanation below. In the revised part, we have made it clear that the ABCDE in Figure 2 corresponds to market access& targets and trade barriers (mar), customers efficiency (cus), information and communication technology (ict), infrastructure construction (inf), and business environment (bus), respectively. We provided detailed explanations in Line 165-167. Furthermore, we also explained their specific connotations in Line 165, 173, 191, 208, 222 and 237.

3. There is need to add the theoretical underpinnings of the study as well, for example different models of trade, pertinent to this study, like, the gravity theory of international trade etc.

Response: Thanks for your suggestions. In the revised part, we completed previous research in the academic community on the gravitational model of international trade before using Quadratic Assignment analysis. And we also explained why QAP analysis method was chosen in this article. (Line 324-339)

4. Fig.6 (depicting SNA results) is fade and does not clearly exhibit the trade related networking among the CAFTA countries. Similarly, there is no explanation about the figure in terms of what the figure is showing through dark and dim multicolored lines in different time periods.

Response: Thanks for valuable comments. We’re sorry that the description of Figure 6 was unclear and lacked depth. In the revised part, we provided a detailed description of the flow of trade value added between countries and its profound significance in regional value chains and trade products in 2001, 2010, and 2016. And we would like to explain that, Fig.6a to 6i represent the three types of trade added value at three time nodes. Therefore, Fig.adg use the same legend, as well as Fig.beh and Fig.cfi, in order to make it easier to compare the changes in the value added of trade in different years. 

5. Table 8 claims to exhibit QAP results of different TFI pillars’ impacts on three value-added in trade networks, whereas only the finding of two trade networks (DVA-INTrex and RDV networks) are there.

Response: Thank you for pointing out this error. We have adjusted the name of Table 8 to QAP results of different TFI pillars' impacts on DVA-INTrex & RDV trade networks. (Line 573)

6. QAP analyses (QAP correlations and QAP regressions for the years 2010 and 2016) are done to investigate the influence of the trade facilitation factors on the intermediate goods trade. Those trade facilitation factors or pillars are: domestic business environment(bus), domestic and foreign market access (mar), customs efficiency (cus), availability and quality of transport infrastructure and services (inf), and information and communication technology (ict). But in the QAP regressions (as evident from the findings given in Table 8), why all the five different pillars of trade facilitation are not included.

Response: Thank you for your suggestion. All the five different pillars of trade facilitation have been included in the QAP regression. In fact, the regression results showed that the domestic business environment (bus) had a significant impact on indirect and transferred value-added in 2010 and 2016. And customers efficiency (cus) and information and communication technology (ict) were significant in 2016. Availability and quality of transport infrastructure and services (inf) and domestic and foreign market access (mar) were not significant. The relevant results correspond to sum-bus, sum-mar, sum-cus, sum-inf, and sum-ict, respectively. And we explained the reason for not being significant in Line 570-572.

7. The findings of the study are discussed least in comparison with the relevant previous literature

Response: Thanks for your valuable comments. According to your and the first reviewer’s suggestion, we tried to add the discussion part of the results. We have added a discussion section-Section 4.3, in which we discussed the results and compared them with relevant previous literature. For example, we compared the research results of trade facilitation on DVA-D, DVA-INTrex, and RDV with previous literature (Line 576-589), and found out that the results showed consistency with previous literature. Moreover, we pointed out that this study further revealed that in regions like CAFTA where tariffs and trade barriers between internal countries are relatively low, the key factor affecting the value of DVA-INTrex and RDV is the business environment. Please see the revised part in Section 4.3 (Line 590-598).

8. Policy implications (section 5.3 of the manuscript) could not be so specific, like stated in the lines 574 and 575: “promote e-commerce and paperless reform, strengthen the individual's Internet skill, eliminate high cost transactions and serious time delays, and reduce administrative corruption”. 

Response: Thanks for your valuable comments. We have removed the original inappropriate statements and rewritten the policy implications section in Line 644-651.

---

## [Decision Letter · Decision Letter 1]

26 Jul 2023

Main drivers of regional value chains in CAFTA: Does trade facilitation matter?

PONE-D-23-06237R1

Dear Dr. Song,

We’re pleased to inform you that your manuscript has been judged scientifically suitable for publication and will be formally accepted for publication once it meets all outstanding technical requirements.

Kind regards,

Muhammad Khalid Bashir, PhD

Academic Editor

PLOS ONE

Additional Editor Comments (optional):

Reviewers' comments:

Reviewer's Responses to Questions

**Comments to the Author**

1. If the authors have adequately addressed your comments raised in a previous round of review and you feel that this manuscript is now acceptable for publication, you may indicate that here to bypass the “Comments to the Author” section, enter your conflict of interest statement in the “Confidential to Editor” section, and submit your "Accept" recommendation.

Reviewer #1: All comments have been addressed

Reviewer #2: All comments have been addressed

2. Is the manuscript technically sound, and do the data support the conclusions?

Reviewer #1: Yes

Reviewer #2: Yes

3. Has the statistical analysis been performed appropriately and rigorously? 

Reviewer #1: Yes

Reviewer #2: Yes

4. Have the authors made all data underlying the findings in their manuscript fully available?

Reviewer #1: Yes

Reviewer #2: Yes

5. Is the manuscript presented in an intelligible fashion and written in standard English?

Reviewer #1: Yes

Reviewer #2: Yes

6. Review Comments to the Author

Reviewer #1: The authors have almost addressed all the comments given during the first review. The article is acceptable. Further, The discussion section has still room for improvement. According to the volume of the results, the authors could still detailed discussion.

Reviewer #2: In response to the previously proposed suggestions, the authors have incorporated all the suggestions. Undoubtedly, the articles is suitable for publications in this current form.

1. Abstract is more comprehensive now

2. The diagram (Fig.2) is explained further.

3. Theoretical underpinnings of the study are improved

4. Fig.6 (depicting SNA results) is further explained

5. Table 8 is improved

6. In QAP analyses different pillars of trade facilitation have been included.

7. The findings of the study are better discussed in comparison with the relevant previous literature

8. Policy implications are improved

7. PLOS authors have the option to publish the peer review history of their article (what does this mean?). If published, this will include your full peer review and any attached files.

Reviewer #1: No

Reviewer #2: No

---

## [Editor Report · Acceptance letter]

15 Aug 2023

PONE-D-23-06237R1 

Main drivers of regional value chains in CAFTA: Does trade facilitation matter? 

Dear Dr. Song:

I'm pleased to inform you that your manuscript has been deemed suitable for publication in PLOS ONE. Congratulations! Your manuscript is now with our production department. 

Kind regards, 

on behalf of

Dr. Muhammad Khalid Bashir 

Academic Editor

PLOS ONE